# The Non-Invasive Detection of Pulmonary Exacerbations in Disorders of Mucociliary Clearance with Breath Analysis: A Systematic Review

**DOI:** 10.3390/jcm13123372

**Published:** 2024-06-07

**Authors:** Emma Nessen, Belle Toussaint, Joël Israëls, Paul Brinkman, Anke-Hilse Maitland-van der Zee, Eric Haarman

**Affiliations:** 1Department of Respiratory Medicine, Amsterdam UMC, 1100 DD Amsterdam, The Netherlands; e.j.nessen@student.vu.nl (E.N.); b.toussaint@student.vu.nl (B.T.);; 2Department of Paediatric Pulmonology, Amsterdam UMC, 1100 DD Amsterdam, The Netherlands

**Keywords:** mucociliary clearance, cystic fibrosis, primary ciliary dyskinesia, exacerbation, breath analysis, volatile organic compounds, exhaled breath condensate, GC-MS, eNose

## Abstract

**Background:** Disorders of mucociliary clearance, such as cystic fibrosis (CF), primary ciliary dyskinesia (PCD) and bronchiectasis of unknown origin, are characterised by periods with increased respiratory symptoms, referred to as pulmonary exacerbations. These exacerbations are hard to predict and associated with lung function decline and the loss of quality of life. To optimise treatment and preserve lung function, there is a need for non-invasive and reliable methods of detection. Breath analysis might be such a method. **Methods:** We systematically reviewed the existing literature on breath analysis to detect pulmonary exacerbations in mucociliary clearance disorders. Extracted data included the study design, technique of measurement, definition of an exacerbation, identified compounds and diagnostic accuracy. **Results:** Out of 244 identified articles, 18 were included in the review. All studies included patients with CF and two also with PCD. Age and the definition of exacerbation differed between the studies. There were five that measured volatile organic compounds (VOCs) in exhaled breath using gas chromatography with mass spectrometry, two using an electronic nose and eleven measured organic compounds in exhaled breath condensate. Most studies showed a significant correlation between pulmonary exacerbations and one or multiple compounds, mainly hydrocarbons and cytokines, but the validation of these results in other studies was lacking. **Conclusions:** The detection of pulmonary exacerbations by the analysis of compounds in exhaled breath seems possible but is not near clinical application due to major differences in results, study design and the definition of an exacerbation. There is a need for larger studies, with a longitudinal design, international accepted definition of an exacerbation and validation of the results in independent cohorts.

## 1. Introduction

Mucociliary clearance is one of the most important barriers to protect the respiratory system from pathogens and toxins. For the optimal effect, this system needs the coordinated function of the ciliated epithelial cells, right amount of airway surface liquid and correct composition of the mucus gel layer [1]. In mucociliary clearance disorders (MCDs), such as cystic fibrosis (CF), primary ciliary dyskinesia (PCD) and bronchiectasis, this system is hampered, resulting in recurrent and chronic infections of the respiratory tract. Patients with mucociliary clearance disorders typically have symptoms of a chronic ‘wet’ cough and dyspnoea, with periods of increased symptoms referred to as pulmonary exacerbations (PEx). These exacerbations are usually triggered by pulmonary infections with bacteria, viruses or fungi, which induce inflammation due to, among others, neutrophils releasing proteases (like elastase) and oxidants. In MCDs, recurring pulmonary exacerbations damage the airways, further hampering mucociliary clearance which increases the odds of future exacerbations. In time, this will lead to structural damage to the airways (bronchiectasis) and a decline in lung function. The early detection of infection and exacerbation is associated with the preservation of lung function and quality of life [2]. Therefore, adequate interventions to halt and prevent exacerbations are necessary to avert tissue damage and preserve quality of life in patients suffering from mucociliary clearance disorders. For this, the early detection of inflammation and pathogens is key.

There is no consensus on the precise definition of a pulmonary exacerbation. In general, an exacerbation is considered a worsening of symptoms, like increased cough or shortness of breath, and a change in clinical parameters, such as lung function [3]. In striving for a universal definition of exacerbations, different organisations and research groups have suggested sets of criteria based on clinical parameters, including symptoms and measurements. For CF, these include the EPIC (Early Pseudomonas Infection Control) criteria as well as the EuroCareCF Working Group definition for exacerbation, following changes in clinical parameters [4,5]. Similarly, for PCD, a consensus on exacerbation definitions was set up by the BEAT-PCD network for use in clinical trials and other research [3]. For non-CF/non-PCD bronchiectasis, a definition of PEx was proposed in a 2021 clinical guideline by the European Respiratory Society (ERS) [6] and for research in a 2022 paper [7]. In symptom-defined exacerbation scores like these ones, variability is likely to be introduced [4]. All sets of criteria for the definition of an exacerbation are rather extensive and open to different interpretations. The need to avoid variability in clinical assessment drives the search for a more objective identification of a PEx and subsequent more adequate treatment [8]. A possible method for detecting pulmonary exacerbations could be the analysis of exhaled breath.

Volatile organic compounds (VOCs) in exhaled breath and organic compounds in exhaled breath condensate (EBC) have been suggested as biomarkers for respiratory inflammation and the presence of pathogens [9]. VOCs are carbon-based molecules with high volatility at room temperature. They are present in exhaled breath and reflect products of organic metabolism, ranging from humans to bacteria [9]. The measurement of VOCs is usually performed with mass spectrometry (MS)-based technology to detect individual VOCs or with pattern recognition-based technology, which is often referred to as an electronic nose (eNose) [10]. The volatile and non-volatile macromolecules in EBC originate mainly from the airway-lining fluid and include multiple markers of inflammation. The method of choice for EBC analysis depends on the compound of interest with multiple possible techniques such as spectrophotometry and immunoassay [11].

In the last decade, there has been an increased interest in the possibility for the detection of pathogens through exhaled breath analysis, for instance in patients with CF. This technique has not been clinically implemented yet, since the validation of results is currently a topic of research within several groups in the world [12]. When steering towards the clinical implementation of breath analysis, it is important to consider the origin of biomarkers. It would be beneficial to differentiate between pathogen-specific biomarkers and inflammation-specific biomarkers, as colonisation with a pathogen is not necessarily indicative of the inflammatory response. The detection of pulmonary exacerbations through an analysis of exhaled metabolites will allow for a less invasive detection of inflammation and, at the same time, avoid inadequate or unnecessary treatment.

This systematic review assesses the potential of VOCs and EBC biomarkers for predicting pulmonary exacerbations in mucociliary clearance disorders based on recent studies in this field. This assessment will provide recommendations for future research on breath analysis.

## 2. Materials and Methods

### 2.1. Data Sources and Search Criteria

A systematic literature search was performed in March 2024 in MEDLINE (PubMed). The search strategy was built with terms for mucociliary clearance disorders (including CF, PCD, bronchiectasis and suppurative lung disease) AND breath analysis (including breath test, volatile organic compound, VOC, GC-MS, eNose, EBC) AND infection or exacerbation. There were no filters applied during the search process.

### 2.2. Selection Process

The title and abstract of all articles were independently reviewed by three reviewers (E.N., B.T. and J.I.) with Rayyan (https://rayyan.qcri.org/ (accessed on 26 March 2024)). We selected clinical studies that explored the diagnostic value of exacerbation detection in CF, PCD and bronchiectasis by measuring VOCs in exhaled breath or non-volatile organic biomarkers in EBC (such as cytokines and purines). Included studies were required to compare clinically stable patients to those experiencing a pulmonary exacerbation. Review articles, meta-analyses, case reports, animal studies and in vitro studies were excluded. Full texts were examined to determine their inclusion in the review. Studies examining non-organic, gaseous compounds in exhaled breath were excluded; these were mainly articles on nitric oxide (NO), hydrogen peroxide (H_2_O_2_), carbon monoxide (CO) and microRNA in EBC. Any disagreement between the reviewers was resolved through discussion.

### 2.3. Data Extraction and Synthesis

Descriptive data were extracted regarding study design, setting, patient characteristics, sampling method, outcome measures, identified VOCs or biomarkers found in EBC and diagnostic accuracy. Of all the variables, only data relevant for the detection of exacerbations were selected. The relevant patient characteristics were the diagnosis, age (paediatric, adult or both) and number of patients with a pulmonary exacerbation (PEx). Additionally, the inclusion criteria from each study were examined. Given the variability in the definitions used to diagnose pulmonary exacerbations across studies, special attention was directed towards extracting this information.

## 3. Results

### 3.1. Study Selection

An overview of the study selection process is presented in Figure 1, according to the PRISMA reporting guidelines. In total, 244 records were compatible with the search terms in MEDLINE (PubMed). Removing two duplicates, 242 records remained, of which titles and abstracts were screened and reviewed. Studies that did not meet the inclusion criteria were excluded, after which 39 articles were assessed for inclusion by the screening of the full text. A total of 18 studies were included for data extraction in this review.

### 3.2. Description of Included Studies

An overview of the characteristics and main findings of the included articles is provided in Table 1. All studies included patients with CF (*n* = 18), and two studies also included patients with PCD. None of the studies included a cohort of patients with bronchiectasis of unknown origin. Most studies were based on a paediatric population (*n* = 16), and in 11 cases, adults were excluded. Four articles reported on exhaled VOCs measured with GC-MS [13,14,15,16], two papers concerned eNose-driven work [17,18] and one manuscript was focused on hydrogen cyanide in exhaled breath [19]. Biomarkers in exhaled breath condensate were measured in 11 studies. Of these eleven, four used ultra-performance liquid chromatography with mass spectrometry (UPLC-MS) to measure VOCs [20,21], asymmetric dimethylarginine (ADMA) and related amino acids [22] and purines like adenosine and AMD [23]. Six articles on EBC used an immunoassay [24,25,26,27,28,29] to measure cytokines (five studies), LTB_4_ (two studies), 8-isoprostane, VEGF, e-Cadherin and neutrophil elastase. In one study, ATP was measured in EBC through luminometry [30]. The studies that focused on VOCs in exhaled breath (GC-MS and eNose) had mainly a longitudinal cohort or case–control design. Here, samples of CF patients were collected at an outpatient clinic and compared, on the basis of clinical status, between a stable versus unstable phase of disease. The eleven studies measuring biomarkers in EBC were mainly designed as cohort studies, of which six had a longitudinal component and measured EBC at the start of treatment for a PEx and at the end of treatment. The remaining five studies were designed as a cross-sectional cohort, measuring EBC in a sample of patients with PCD and comparing biomarkers in those with a PEx to those without. The sample size of the studies was between 12 and 138 patients, with an average of around 40 enrolments. Larger cohorts were mostly in studies using EBC. Those measuring VOCs with GC-MS in exhaled breath included 12 to 49 patients. The percentage of patients with a pulmonary exacerbation differed from 14 out of 85 [18] to 32 out of 49 [28] and was evidently 100% in the longitudinal studies where patients were included during an exacerbation.

### 3.3. The Definition of an Exacerbation

The definition of a pulmonary exacerbation differed between most of the included studies and was not mentioned in one study [19]. An overview of these definitions is presented in Table 2. In eight studies, the clinical decision to start therapeutic antibiotics was one of the criteria and for three of those eight studies, the only criterion. Most definitions combined an increase in respiratory symptoms, >10% decrease in FEV1 and radiographic changes to define a PEx, either as ‘conventional criteria’ or according to different proposed criteria (e.g., EPIC trial criteria, EuroCareCF working group definition). Signs of infection (fever, leukocytosis, CRP) were incorporated in the definition by three studies. Finally, in six of the included studies, it was not clear which combination of the criteria was needed to define a pulmonary exacerbation.

### 3.4. Exacerbation Detection with VOCs in Exhaled Breath Using Mass Spectrometry

The compounds of interest and main results of the five studies using mass spectrometry on exhaled breath varied. Two studies used a targeted approach for either hydrogen cyanide or isoprene. Hydrogen cyanide levels did not differ significantly between CF patients with and without a PEx [19]; none of the other studies focused on this compound. Isoprene increased significantly from the start of an exacerbation to the end of a two-week intravenous antibiotic treatment [13]. Out of the three studies measuring multiple VOCs, one found pentane to be increased in CF patients with a PEx [14], while other VOCs classified as inflammatory mediators did not differ (such as isoprene, ethanol, propane and dimethyl sulphide). The two remaining studies tried to model VOCs for the detection of a PEx. A model with only 3,7-dimethyldecane showed an area under the receiver operating characteristic curve (AUROCC) of 0.91 [16]. The study with a model of nine VOCs (four hydrocarbons, two aromatic compounds, camphene, tetradecanal and 3-methyl-2-butanone) showed a sensitivity of 79%, specificity of 78% and AUROCC of 0.88 for a pulmonary exacerbation in patients with CF [15]. None of these studies were able to confirm previous findings.

### 3.5. Exacerbation Detection with VOCs in Exhaled Breath Using eNose

The two studies using an electronic nose both used the same device (Cyranose 320, Smiths Detection, Pasadena, CA, USA), with an array of 32 sensors [17,18]. In a paediatric cohort, the eNose showed an AUROCC of 0.76 for a pulmonary exacerbation in patients with CF and 0.90 in patients with PCD [17]. In both patient groups, sensitivity was higher than specificity (89–100% vs. 56–90%). In the cohort of both adults and children, the eNose was able to detect a PEx with an AUROCC of 0.69 in patients with CF (sensitivity 90%, specificity 50%), while no differences were found for patients with PCD [18].

### 3.6. Exacerbation Detection with Biomarkers in Exhaled Breath Condensate

Multiple VOCs in exhaled breath condensate were correlated with the presence of a pulmonary exacerbation. In both studies by Zang et al., 4-hydroxycyclohexylcarboxylic acid and pyroglutamic acid were increased in, respectively, adult and paediatric CF patients with a PEx [20,21]. Most results differed between adults and children; the only overlapping outcomes were increased C_9_H_10_O_3_ (not further identified) and lowered acetic acid. None of the previous mentioned VOCs, found with MS in exhaled breath and correlated to a PEx, were reported to correlate with an exacerbation in the EBC analysis. Significant models incorporating cytokines IL-6, IL-8, TNF- α and MIF [28] or 8-isoprostane and nitrite [26] were calculated in the largest studies (49 and 48 included patients) measuring EBC in a cross-section cohort of children with CF. The confirmation of the relationship between a PEx and the level of certain cytokines (mainly IL-6, IL-8 and TNF-α) was positive and negative in two studies each. The same goes for Leukotriene B4 (LTB_4_) where the replication of a possible relation with PEx was negative [29]. When measuring purines in EBC, one study showed a significant decrease in ATP after the treatment of a PEx. This finding could not be confirmed in a slightly larger cohort of the same study group [23]. Finally, no significant differences could be found when measuring asymmetric dimethylarginine and related amino acids in EBC [22].

## 4. Discussion

In this review, we show multiple compounds of interest in exhaled breath that could help to identify a pulmonary exacerbation in patients with cystic fibrosis. For other disorders of mucociliary clearance, such as bronchiectasis of unknown origin or primary ciliary dyskinesia, no such conclusions can be drawn due to a lack of studies. Hydrocarbons, such as alkenes, seem the most promising when exhaled breath is measured using GC-MS, showing a significant relation in four of the included studies. The same goes for cytokines in exhaled breath condensate, which were significantly correlated to a PEx in three out of five included studies. When focusing on single compounds, such as pentane or IL-6, there was a lack of consistent results between the studies. Therefore, it is not yet possible to conclude which technique or specific compound holds the future for the non-invasive detection of pulmonary exacerbations.

The strength of our review lies in the systematic assessment of all included studies according to the PRISMA checklist [31] and broad literature search to include as many relevant articles as possible. No specific terms for other forms of spectrometry (such as proton-transfer reaction MS or GC with ion mobility spectrometry) were included in our search strategy. We presume relevant articles with these techniques would also have shown up in our search, based on the other included terms on breath analysis (such as breath test, exhaled breath or VOC). For the sake of focus, we decided not to include studies on gaseous or non-volatile compounds. Therefore, we cannot make a conclusion on the potential to detect pulmonary exacerbations with the measurement of, among others, exhaled nitrous oxide, carbon monoxide, hydrogen peroxide, the pH of EBC or the presence of microRNA in EBC.

The results of most of the included studies were contradicting. When evaluating these results, it is important to consider the heterogeneity of the studies. There were major differences in study design, including cross-sectional versus longitudinal cohorts and the included age group. In the cross-sectional design, the number of pulmonary exacerbations is based on chance, and, with small sample sizes, the absolute number of exacerbations is minimal. Specifically selecting patients during an exacerbation increases those numbers but introduces significant selection bias. The results of studies with this design might represent compounds related to high-risk patients, instead of the exacerbation itself. The included studies with a longitudinal design had the same form of bias because patients were included during an exacerbation in all. Furthermore, in most of these studies, the measurements were performed at the start and end of treatment for a PEx. The start of treatment is during a period of inflammation, but we cannot be sure about the complete resolution of inflammation at the end of treatment. So, ideally, sample sizes should be large enough to include enough exacerbations. In a longitudinal design, inclusion should start cross-sectionally, or enough time should pass between the start and resolution of the exacerbation. The included age group has a significant effect on the different confounding variables, such as bacterial colonisation or, in cystic fibrosis, CF-related diabetes. Studies excluding adults therefore seem the most promising with less bias by these kinds of variables. When all ages are included, separate analyses in adults and children are preferred.

All studies included patients with cystic fibrosis, and therefore, the year of inclusion also effects the results. Since the introduction of highly effective modulator therapy (HEMT), the course of this disease is seriously altered for most of the patients who are on this medication. Prognosis is greatly improved, and exacerbations are much less frequent. Breath profiles are also influenced by HEMT, as shown in the results of Woollam et al., where patients not taking HEMT had increased octanal and nonanal in exhaled breath [16]. The results before and after the introduction of HEMT are therefore difficult to compare. Most of the included studies were conducted before the introduction of HEMT. Only one article mentioned the number of patients using HEMT [16]. Thus, the current results mainly represent a CF population without access to HEMT. For future research, new datasets with CF patients using HEMT are required.

No single VOC or biomarker in exhaled breath condensate seems to be able to accurately diagnose the presence of a pulmonary exacerbation. Although some of the included studies report on a specific compound (e.g., pentane or IL-8), there is lack of a successful replication of these single findings in this field of research. Out of all the included studies, the best results were found when combining multiple compounds or measuring a VOC profile with an electronic nose. These strategies therefore seem to be the most promising for future care. To select the most relevant compounds or sensors, there is still a need for prospective studies that target specific compounds of interest, in other words potential biomarkers.

The origin of the compound of interest will probably affect its meaning as a predictor. Multiple hydrocarbons, aromatic compounds, pentane and isoprene, present as VOCs in exhaled breath, were all described to be related to a pulmonary exacerbation [13,14,15,16]. Some of these compounds are known to be associated with the presence of specific microorganisms, such as camphene with Aspergillus fumigatus and pentadecane with Pseudomonas aeruginosa and Staphylococcus aureus [12]. Others, such as the hydrocarbons pentane and isoprene, are more closely related to inflammation [13,14]. These compounds are formed during oxidative stress with the generation and interaction of reactive oxygen species by inflammatory cells. Although all of these compounds might be associated with an exacerbation, the former will be influenced by the bacterial colonisation of the patient, and the latter might be more specific for the inflammation during an exacerbation. When creating a model out of multiple compounds, it is therefore preferred to mainly include compounds associated with inflammation or to validate the model in a cohort of patients with different chronic infections. Most of the measured biomarkers in EBC are closely related to inflammation and might be more suitable to predict an exacerbation. Still, none of the measured cytokines in EBC showed consistent results between the studies. Of these, IL-6 and IL-8 seem the most promising due to their biologic pathway and positive result in three out of five studies, including the largest sample size to date [28]. The two studies that could not correlate these cytokines to a pulmonary exacerbation suffered from either a small study group of 24 patients >12 years of age [27] or did not perform measurements during an exacerbation but compared children with and without an exacerbation in the previous year [29].

## 5. Conclusions

A summary of the main findings and advice for future research is presented in Table 3. In conclusion, this review shows the potential of breath analysis for the detection of pulmonary exacerbations in disorders of mucociliary clearance. Significant correlations were found for multiple biomarkers present as VOCs in exhaled breath or organic compounds in exhaled breath condensate. Most potential seems to reside in models with multiple compounds, including hydrocarbons and cytokines.

For clinical care, we need a real-time form of measurement. As most potential lies in the analysis of a combination of compounds, real-time pattern recognition, such as the electronic nose technique, seems the most suitable for future implementation. In future research, we need to discover which combination of compounds is the most predictive of a pulmonary exacerbation. To that end, we need to study multiple single compounds, associated with a PEx, and combine these in prediction models. Studies should use a uniform definition of a PEx and include larger cohorts with a longitudinal design, increasing the odds of an exacerbation. Positive results should be validated in separate cohorts to reduce selection bias.

## Figures and Tables

**Figure 1 jcm-13-03372-f001:**
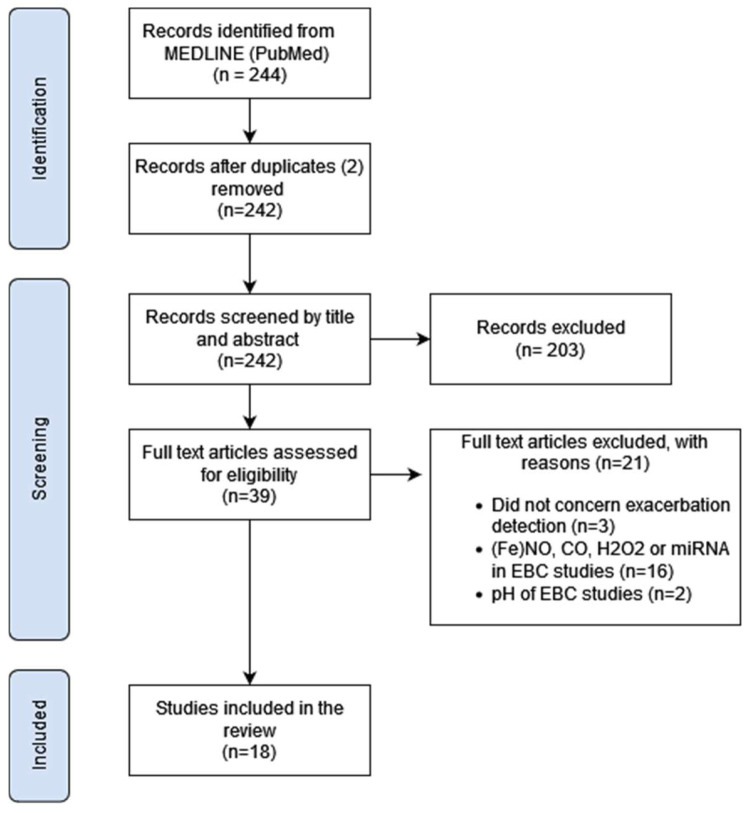
Flowchart of study selection. Abbr: (Fe)NO: (fractional exhaled) nitric oxide, CO: carbon monoxide, H_2_O_2_: hydrogen peroxide, miRNA: microRNA, EBC: exhaled breath condensate.

**Table 1 jcm-13-03372-t001:** Overview of study characteristics of selected papers.

**Part 1: VOCs in exhaled breath measured with GC-MS**							
		**Study population**				**Results**	
**Author, year**	**Design**	**Disease**	**Adult/Paediatric**	**Total** **[*n* PEx]**	**Technique**	**Biomarkers included in the model**	**Main result**
McGrath, L.T., 2000 [13]	Longitudinal cohort	CF	Adult	12 (12)	GC-MS	Isoprene	Significant increase after treatment of PEx
Barker, M., 2006 [14]	Case–control: CF with vs. without PEx	CF	Both	20 (5)	GC-MS	Pentane	Significantly increased during Pex
Enderby, B., 2009 [19]	Longitudinal cohort	CF	Paediatric (≥7 years)	16 (not stated)	SIFT-MS	Hydrogen Cyanide	No significant changes during PEx
Van Horck, M., 2021 [15]	Longitudinal cohort	CF	Paediatric	49 (31)	GC-tof-MS	Hydrocarbons: C_8_H_18_, C_9_H_20_, 2,4-dimethyl-1-heptene, pentadecane. Aromatic compounds: 1,3-dimethylbenene, p-benzoquinone. Other: Camphene, Tetradecanal, 3-methyl-2-butanone	Sens: 79% Spec: 78% AUROCC: 0.88
Woollam, M., 2022 [16]	Cross-sectional cohort	CF	Paediatric (>8 years)	18 (7)	SPME GC-MS	Hydrocarbon: 3,7–dimethyldecane	Sens: 100% Spec: 73% AUROCC: 0.91
**Part 2: VOCs in exhaled breath measured with eNose**							
		**Study population**				**Results**	
**First author, year**	**Design**	**Disease**	**Adult/Paediatric**	**Total [*n* PEx]**	**Technique**	**Main result**	**Main result**
Paff, T., 2013 [17]	Cross-sectional case–control	CF and PCD	Paediatric	50 (13)	eNose	CF (*n* = 25) Sens: 89% Spec: 56% AUROCC: 0.76	PCD (*n* = 25) Sens: 100% Spec: 90% AUROCC: 0.90
Joensen, O., 2014 [18]	Cross-sectional case–control	CF and PCD	Both	85 (14)	eNose	CF (*n* = 64) Sens: 90% Spec: 50% AUROCC: 0.69	PCD (*n* = 21) No significant differences found
**Part 3: Biomarkers in exhaled breath condensate**							
		**Study population**				**Results**	
**First author, year**	**Design**	**Disease**	**Adult/Paediatric**	**Total [*n* PEx]**	**Technique**	**Biomarkers included in the model**	**Main result**
Carpagnano, G.E., 2003 [24]	Longitudinal cohort	CF	Adult	20 (20)	Immunoassay	LTB_4_ and IL-6	Significant decrease after treatment of PEx
Bodini, A., 2007 [25]	Longitudinal cohort	CF	Paediatric	15 (15)	Immunoassay	IL-8	Significant decrease after treatment of PEx
Robroeks, CM., 2008 [26]	Cross-sectional cohort	CF	Paediatric	48 (6)	Immunoassay	8-isoprostane and nitrite	Sens: 40% Spec: 97% AUROCC: 0.84
Esther, C.R., 2008 [30]	Longitudinal cohort	CF	Paediatric	14 (14)	Luminometry	ATP	Significant decrease after treatment of PEx
Esther, C.R., 2009 [23]	Longitudinal cohort	CF	Paediatric	26 (26)	UPLC-MS	Purine to urea ratio	No significant differences
Colombo, C., 2011 [27]	Longitudinal cohort	CF	Both (>12 years)	24 (24)	Immunoassay	Cytokines and growth factors (e.g., IL-6, IL-8, IL-10, TNF-α, VEGF, IFN-y)	No significant differences
Van Horck, M., 2016 [28]	Cross-sectional cohort	CF	Paediatric	49 (32)	Immunoassay	IL-6, IL-8, TNF-α, MIF	Sens: 70% Spec: 50% AUROCC: 0.62
Zang, X., 2017 [20]	Cross-sectional cohort	CF	Both	26 (9)	UPLC-MS	Pyroglutamic acid and 4-hydroxycyclohexylcarboxylic acid	Sens: 77.8% Spec: 88.2% Accuracy 84.6%
Lucca, F., 2018 [22]	Longitudinal cohort	CF	Paediatric	34 (13)	UPLC-MS	Asymmetric dimethylarginine (ADMA) and related amino acids	No significant differences
Zang, X., 2020 [21]	Cross-sectional cohort	CF	Both	138 (41)	UPLC-MS	*Paediatric patients* Higher in PEx: lactic acid, pyroglutamic acid, dihydrothymine, C_5_H_9_NO_3_, prolylhydroxyproline, C_9_H_10_O_3_ and C_7_H_8_O_4_S Lower in PEx: acetic acid and 3-methylglutaconic acid *Adult patients* Higher in PEx: 4-hydroxy-cyclohexylcarboxylic acid, nonanedioic acid, sebacic acid, y-butyrolactone, levulinic acid, C_9_H_10_O_3_ and C_6_H_10_O_6_ Lower in PEx: acetic acid and C_8_H_11_NO_3_	Sens: 83.3% Spec: 91.7% Accuracy 88.9% Sens: 76.2% Spec: 83.7% Accuracy 81.3%
Toprak, Kanik E., 2020 [29]	Case–control: CF with and without PEx in past year	CF	Paediatric	30 (10)	Immunoassay	IL-8, IL-17, LTB4, E-cadherin and neutrophil elastase	No significant differences

Abbr: AUROCC: area under the receiver operating characteristic curve; PEx: pulmonary exacerbation; CF: cystic fibrosis; PCD: primary ciliary dyskinesia; GC-MS: gas chromatography with mass spectrometry; tof: time of flight; SPME: solid phase microextraction; UPLC-MS: ultra-performance liquid chromatography with mass spectrometry.

**Table 2 jcm-13-03372-t002:** An overview of the used definition or criteria for a pulmonary exacerbation across studies.

Author, Year	The Definition of a Pulmonary Exacerbation
McGrath, L.T., 2000 [13]	Increase in respiratory symptoms AND >10% decrease in FEV1 compared to previous year AND Decision to treat with intravenous antibiotics
Carpagnano, G.E., 2003 [24]	Increase in respiratory symptoms >10% decrease in FEV1 compared to previous year Signs of infection (fever, increase CRP or leukocytosis) Bacterial colonisation of sputum
Barker, M., 2006 [14]	Opinion of clinician: PEx in need of intravenous antibiotics
Bodini, A., 2007 [25]	‘Conventional criteria’: clinical symptoms, radiology, >10% decrease in FEV1, increased CRP and leukocytosis
Robroeks, CM., 2008 [26]	Increase in respiratory symptoms AND/OR >10% decrease in FEV1 or FVC from baseline
Esther, C.R., 2008 [30]	Opinion of clinician: PEx in need of intravenous antibiotics
Esther, C.R., 2009 [23]	
Enderby, B., 2009 [19]	Not defined in the article
Colombo, C., 2011 [27]	Increase in respiratory symptoms Decrease in FEV1 compared with previous best Weight loss and fever
Paff, T., 2013 [17]	Additional antibiotic treatment due to respiratory symptoms, >10% decrease in pulmonary function or radiographic changes
Joensen, O., 2014 [18]	Additional antibiotic treatment due to respiratory symptoms, >10% decrease in pulmonary function or radiographic changes
Zang, X., 2017 [20]	Increase in respiratory symptoms and/or changes in physical examination of the lungs >10% decrease in FEV1 According to clinician in need of hospitalisation for treatment of PEx
Zang, X., 2020 [21]	
van Horck, M., 2016 [28]	EPIC trial criteria: ≥5 days of respiratory symptoms, >10% decrease in FEV1, radiographic changes AND/OR Opinion of clinician: PEx in need of therapeutic antibiotics
van Horck, M., 2021 [15]	
Lucca, F., 2018 [22]	Definition of EuroCareCF working group. Two of the following: Respiratory symptoms OR >10% decrease in FEV1 OR Radiographic changes
Toprak Kanik, E., 2020 [29]	Increase in respiratory symptoms or signs Radiographic changes >10% decrease in spirometry
Woollam, M., 2022 [16]	Opinion of clinician: PEx in need of therapeutic antibiotics AND/OR >10% decrease in FEV1 from baseline

**Table 3 jcm-13-03372-t003:** Overview of main findings and advice for future research.

**Main findings**
- Hydrocarbons and alkenes in exhaled breath seem the most promising for detecting a PEx. - Cytokines (such as IL-6 and IL-8) in EBC have potential for the detection of a PEx. - eNose technology was able to detect a PEx in exhaled breath with a sensitivity of 89–100% and specificity of 50–90%. - The lack of consistent results might be caused by differences in the definition of a PEx, study design, study group (paediatric vs. adult), study period (before vs. after introduction of HEMT) and small sample sizes.
**Advice for future research**
- A uniform definition of a PEx. - Larger cohorts, preferably with a longitudinal design. - Separate analyses of adults and children. - Combine multiple compounds of interest into a model. - The validation of results in different cohorts.

## Data Availability

Data sharing is not applicable.

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
