# Peer review of "The Non-Invasive Detection of Pulmonary Exacerbations in Disorders of Mucociliary Clearance with Breath Analysis: A Systematic Review"

_jcm, 2024, doi:10.3390/jcm13123372_

Round 1

Reviewer 1 Report

Comments and Suggestions for Authors

Title - Non-invasive detection of pulmonary exacerbations in disorders of mucociliary clearance with breath analysis: a systematic review

This systematic review is well written with good English and no obvious grammatical errors

244 potentially appropriate studies were identified but only 18 included in review after screening

All 18 papers concerned CF patients but only 2 of these studies examined PCD patients.

Authors provide detailed description of studies and Table 1 good

Agree shows the potential of breath analysis for the detection of pulmonary exacerbations in CF but more studies needed and need better definitions for PEx

Issues to be addressed:

This review essentially a breath analysis study of CF exacerbations, and predominantly pediatric cohort (16/18).  Need to take out discussion re bronchiectasis of unknown origin as no studies used included these patients and explain clearly why relevant to compare CF and PCD (only 2 studies for later and different disease)

Address fact that updated CF studies needed in patients on HEMT – not relevant to >90% current CF population

Main problem is that review is comparing heterogenous studies. probably no way round this but should be addressed in discussion

Reviewer 2 Report

Comments and Suggestions for Authors

Nessen et al. evaluated in their review of existing literature on breath analysis to detect pulmonary exacerbation in patients with CF and/or PTD. A total of 18 articles were included until March 2024. Based on the reports, significant discrepancies were found in the criteria for inclusion of patients and the definition of pulmonary exacerbation, which were analyzed in detail. The review is brief and substantive, including data from the latest clinical trials. The literature used is current and accurately quoted. The summaries and conclusions are correct and can serve as guidance later on.

Author Response

We would like to thank you for the positive feedback and your time and effort to review the manuscript.

Reviewer 3 Report

Comments and Suggestions for Authors

Systematic Review: Non-invasive detection of pulmonary exacerbations in disorders of mucociliary clearance with breath analysis: a systematic review

This is an orderly, clear review and search for exhaled organic compounds that may be possible biomarkers of serious and frequently high lung conditions such cystic fibrosis and others such as mucociliary clearance disorders.

Is necessary remember that patients with mucociliary clearance disorders typically have symptoms of a chronic ‘wet’ cough and dyspnea, with periods of increased symptoms referred to as pulmonary exacerbations. These exacerbations are usually triggered by pulmonary infections which induce inflammation.

One of the points that would require a review would be the integration of the compounds that are exhaled and the relationship with the metabolic pathways or the places (cells or tissues) from which they come, would help to advance this topic.

Could you propose what they would be from your point of view or from the review carried out, which markers are possible for you and for which cases?

Other points: 

Check for the presence of unclear abbreviations in the abstract: CF and PCD

Comments on the Quality of English Language

English language is required only minor revision. 
